

# 1 Toward high-spatial resolution hydrological modeling

# 2 for China: Calibrating the VIC model

**Bowen Zhu[1,2], Xianhong Xie[1,2*], Chuiyu Lu[3], Shanshan Meng[1,2], Yi Yao[1,2], Yibing**
**Wang[1,2]**
1. State Key Laboratory of Remote Sensing Science, Jointly Sponsored by Beijing
Normal University and Institute of Remote Sensing and Digital Earth of Chinese
Academy of Sciences, Beijing 100875, China
2. Beijing Engineering Research Center for Global Land Remote Sensing Products,
Institute of Remote Sensing Science and Engineering, Faculty of Geographical
Science, Beijing Normal University, Beijing 100875, China
3. China Institute of Water Resources and Hydropower Research, State Key Laboratory
of Simulation and Regulation of Water Cycle in River Basin, Beijing 100038, China
*Corresponding author: Xianhong Xie (Beijing Normal University,
xianhong@bnu.edu.cn)



**Abstract**
High-resolution hydrological modeling is important for understanding fundamental
terrestrial processes associated with the effects of climate variability and human
activities on water resources availability. However, the spatial resolution of current
hydrological modeling studies is mostly constrained to a relative coarse resolution
(~10–100 km) and they are therefore unable to address many of the water-related issues
facing society. In this study, a high resolution (0.0625º, ~6 km) hydrological modeling
for China was developed based on the Variable Infiltration Capacity (VIC) model,
spanning the period from January of 1970 to June of 2016. Distinct from other modeling
studies, the parameters in the VIC model were updated using newly developed soil and
vegetation datasets, and an effective parameter estimation scheme was used to transfer
parameters from gauged to ungauged basins. Simulated runoff, evapotranspiration (ET),
and soil moisture (SM) were extensively evaluated using in-situ observations, which
indicated that there was a great improvement due to the updated model parameters. The
spatial and temporal distributions of simulated ET and SM were also consistent with
remote sensing retrievals. Moreover, this high-resolution modeling is capable of
capturing flood and drought events with respect to their timing, duration, and spatial
extent. This study shows that the hydrological datasets produced from this high-
resolution modeling are useful for understanding long-term climate change and water
resource security. It also has great potential for coupling with the China Land Data
Simulation System to achieve real-time hydrological forecasts across China.



## 1 Introduction

Climate change and human activities impart substantial influences on hydrological
cycles and water resources, resulting in many challenges in multi-scale hydrological
research (Devia et al., 2015). Water-related research has largely been reshaped by the
need to solve practical problems, such as predicting floods and droughts, managing
water resources, and designing water supply infrastructures at finer scales (Kirchner,
2006). As an alternative solution, high-resolution hydrological modeling is key to
supporting analyses of land–atmosphere interactions, surface and subsurface
interactions, water quality, and human impacts on the terrestrial water cycle (Wood et
al., 2011) , and can serve as a benchmark for evaluating extreme events and for
preventing record-setting disasters in advance (Lee et al., 2017). Developing a high-
resolution hydrological modeling is also recognized as important for understanding the
implications of climate change (Zhu and Lettenmaier, 2007) and improving the ability
of scientists to narrow uncertainties and errors in water resources management (Scherer
et al., 2015).
At present, hydrological modeling are usually implemented at resolutions from 0.125º
to 2º latitude by longitude and with temporal resolutions from hourly to daily
(Cherkauer et al., 2003; Troy et al., 2008) across different regions, such as Mexico (Zhu
and Lettenmaier, 2007), Texas (Lee et al., 2017), and the Mississippi watershed
(Scherer et al., 2015). China is one of the most interesting study areas for many
researchers and hydrological modeling of China have also been simulated in a variety
of studies (Wang et al., 2011; Wang et al., 2012; Xie et al., 2007; Zhang et al., 2014).



However, many terrestrial hydrologic and vegetative states and fluxes are typically
constrained at rather coarse spatial resolutions (~10–100 km), which cannot adequately
address critical scientific questions about the water cycle (Wood et al., 2011) or to
describe hydrological process and water dynamics in small watershed, especially when
there is a need to detect the impact of extreme events.
In recent decades, many devastating natural disasters have occurred frequently
worldwide and in China due to global climate change (Mo et al., 2016; Piao et al., 2010;
Xu et al., 2015). The intensification of droughts and floods is having a critical negative
impact (i.e., economic losses, agricultural destruction) in China (Zhang et al., 2015).
Therefore, high-resolution hydrological modeling in China is urgently needed to
identify and monitor the underlying processes and intensities of hydrological extremes
(Dong et al., 2011) and to reflect the regional details of climate change patterns (Zhang
et al., 2006).
However, there are disadvantages and difficulties in developing high-resolution
hydrological modeling in China with respect to meteorological forcings, soil and
vegetation datasets, and model evaluation. First, meteorological forcing data hold
substantial uncertainties, especially for high-resolution modeling, because ground-
based observation stations are limited in China. Only ~750 meteorological stations for
collecting data (which may be combined with remote sensing datasets) have been
commonly used to generate different resolutions of forcing data (Xu et al., 2015; Zhai
et al., 2005; Zhang et al., 2014), and these datasets are only suitable for modeling at
coarse resolutions (> 10 km) rather than at high-resolutions. Second, estimating model





parameters presents a great challenge because the climate, soil, and land cover
conditions are highly heterogeneous over the 9.6 million km$^2$ area of China (Zhai et al.,
2005). Third, ground-based hydrological stations are extremely scarce in most basins,
and hydrological datasets are insufficient for model calibration and validation. Thus, in
many studies, model parameters have only be calibrated using limited streamflow data,
while evapotranspiration (ET) and soil moisture (SM) states have not been well
evaluated (Jiao et al., 2017; Scherer et al., 2015). Finally, remote sensing (RS) data can
serve as hydrological model inputs; however, RS data have not been fully combined
with hydrological modeling, although they have the potential to improve model
performance (Wu et al., 2014).
In this study, we attempt to develop a high-resolution hydrological modeling framework
for China at the spatial resolution of 0.0625º (~6 km). The framework is based on a land
surface hydrological model, (i.e., the Variable Infiltration Capacity (VIC)) (Liang, 1994,
1996). The features of this framework include: (1) it is driven by meteorological forcing
data that were generated based on data from relatively high-density ground-based
stations (2481 stations), nearly tripling the number of meteorological stations when
compared with other studies (Pan et al., 2012; Xie et al., 2007; Zhang et al., 2014); (2)
soil parameters of the VIC were updated based on a newly developed soil dataset for
China, which provides an improved representation of hydrological and biogeochemical
characteristics (Dai et al., 2013; Shangguan et al., 2013); (3) an effective scheme was
employed to estimate model parameters for ungauged basins; (4) the simulated runoff,
ET, and SM were extensively evaluated using ground-based measurements and RS data



products.
This high-resolution modeling framework has attractive applications and potential
extensions. The simulated hydrological flux and state variables are useful for
understanding long-term climatic changes and water resource security at various scales.
Additionally, these simulated variables, benefiting from the high resolution of the
modeling, can provide more detailed information for detecting drought and flood events
at the regional scale. Furthermore, the framework can be extended to couple with the
China Land Data Simulation System (CLDAS), which provides real-time
meteorological inputs and SM conditions at the same resolution (0.0625º) (Shi et al.,
2011). This hydrological modeling, driven by high-quality and real-time inputs from
CLDAS, may improve the accuracy of results and estimate real-time hydrological
processes.
In the next section we describe the structure of the VIC model, including its inputs data
and parameters. The method of calibration and transfer parameters is also presented. In
section 3, we describe the evaluation of model performance over China and the
application of the modeling on extreme events. We discuss its reliability, and potential
and limitation in section 4, and in section 5 we present our conclusions and thoughts on
future directions.
**2. Data and methods**
**2.1 Hydrological model**
The VIC model is a distributed and physically based model that solves for the surface
energy and water balance (Liang, 1994, 1996). Variable Infiltration Capacity model



simulates SM, ET, snow pack, surface runoff, baseflow, and other hydrological
variables in daily or sub-daily time steps. Each grid cell is partitioned into multiple
vegetation types, and the soil column has three soil layers, where each layer
characterizes the dynamic response of the soil to climatic conditions. The VIC model
characterizes multiple land cover types, with one type of bare soil. Each vegetation type
has a leaf area index (LAI), minimum stomatal resistance, roughness length,
displacement length, and relative fraction of the root (Umair et al., 2018).
This model is selected for use in this study due to three main advantages: (1) a simple
conceptual rainfall–runoff model is used that allows the spatial representation of
gridded topography, infiltration rate, soil properties, climate variables, and land covers,
which are important factors in modeling runoff under spatially heterogeneous
conditions (Tesemma et al., 2015); (2) both infiltration and saturation excess runoff
generation mechanisms are considered in the model, making it suitable for application
to both arid and humid regions; (3) simulations of snow and frozen soil processes,
which are necessary for the Tibet Plateau, can be performed. Finally, the VIC model
has also been shown to represent land surface hydrologic processes well in numerous
studies (Luo et al., 2013; Wu et al., 2014), and has been used from global (Bart Nijssen
et al., 2001; Haddeland et al., 2007) to river basin scales (Liang and Xie, 2001) to assess
water resources, land–atmosphere interactions, and overall hydrological budgets.
**2.2 Data for model inputs**
**2.2.1 Meteorological forcing data**
The VIC model is driven by historical meteorological forcing, including precipitation





(mm), minimum and maximum temperature (℃), and wind speed (m/s). We ran the
model in daily time steps from 1970–2016 in a water-balance mode. All of the forcing
data were produced by interpolating ground-based observations from 2481
meteorological stations in China (Fig. 1a), which were obtained from the China
Meteorological Administration (CMA). These data were interpolated into a gridded
dataset (at a resolution of 0.0625º × 0.0625º) by a linear interpolation method using
an inverse squared distance between the stations. At least five stations around the target
grid were searched to conduct this interpolation. A lapse rate of $-6.5℃$ $km^{-1}$ with
respect to the elevation difference between the station and the target grid was used to
reflect the decrease in temperature with increasing elevation. The same interpolation
method for generating gridded forcing data has been successfully applied in previous
VIC simulations(Xie and Cui, 2011; Xie et al., 2015; Xie et al., 2007).
**2.2.2 Vegetation dataset**
Vegetation data needed for VIC simulations included land cover (LC) types and
associated vegetation parameters. Details of the LC types were originally created by
merging a number of Land Satellite Thematic Mapper (Landsat TM) images (Liu et al.,
2010), with a spatial resolution of 1 km. There were 12 types of LC distributed across
China. Based on these LC types, the fractional area of each vegetation type in a grid
cell was calculated.
The parameters for each type of vegetation (e.g., the architectural resistance) are
available from ftp://ftp.hydro.washington.edu/pub/HYDRO/models/VIC/Veg/veg_lib,
except for the LAI. The LAI reflects the amount of available leaf material, and thus,



represents the canopy density and growth of vegetation, and influences the ET process
(Hanes and Schwartz, 2010). Monthly LAI data at a spatial resolution of 0.1º (∼8 km)
were obtained from the Advanced Very High Resolution Radiometer (AVHRR)
satellites acquired between January of 1982 and December of 2006, and they were
derived from an 8-km composited AVHRR Normalized Difference Vegetation Index
(NDVI) (Strahler et al., 1999). Hence, based on the LC maps and the LAI data,
vegetation parameters were generated for use in the VIC simulations.
**2.2.3 Soil dataset**
Soil datasets define the soil physical and chemical properties of grid cells. In this study,
detailed information on the physical and chemical properties of soils were obtained
based on a 30 × 30 arc-second-resolution soil characteristics dataset (Dai et al., 2013;
Shangguan et al., 2013) derived by using the 1:1,000,000 Soil Map of China and 8595
representative soil profiles.
This dataset is specifically suitable for land surface modeling, so it can be incorporated
into hydrological models to better represent the role of soils in hydrological and
biogeochemical cycles in China. Four influential soil parameters (i.e., field capacity,
wilting point, saturated hydraulic conductivity, and bulk density) for each of the three
layers were obtained from the soil dataset (Dai et al., 2013; Shangguan et al., 2013) and
then applied to the 0.0625º grid in this study. The other soil parameters, such as the
thermal damping depth, bubbling pressure, surface roughness of bare soil, and
snowpack were prescribed according to the Food and Agriculture Organization (FAO)
of the United Nations (UN) dataset, which has been successfully used by Nijssen et al.





(2000); Nijssen et al. (1999).

### 2.3 Data for model evaluation

### 2.3.1 Streamflow

The VIC model was first calibrated and validated using streamflow data. We obtained
streamflow data for 29 stations from the Annual Hydrological Report for P.R. China
(Fig. 1b). These stations are situated at the outlets of 29 sub-catchments that have
different climatic and LC conditions. The data were partitioned into two groups, 20
stations of data were used for calibration and the remaining 9 were then used for model
validation.

### 2.3.2 Evapotranspiration (ET)

Evapotranspiration is the second largest term in the global land surface water budget
(Bohn and Vivoni, 2016), and it was evaluated in our model by using ground-based
observations and an RS product. Ground-based observations of ET were obtained at 33
covariance tower stations (Fig. 1b). The RS ET product was from the Global Land
Surface Satellite (GLASS) and, which merges multiple sources of RS data to achieve
reliable ET estimates (Liang et al., 2013; Yao et al., 2015; Yao et al., 2014; Zhao et al.,
2013), and thus it was used to spatially evaluate the VIC-simulated ET. Moreover, the
GLASS ET was approximately equal in spatial resolution (0.05º) to the model in this
study, and was therefore applicable to the evaluation of ET.

### 2.3.3 Soil moisture (SM)

Soil moisture plays an important role in the terrestrial hydrological cycle, and it also
connects agricultural drought events. Therefore, the validation of SM was also



performed in this study. Soil moisture data from 45 in situ stations across China (Fig.
1b) obtained from the CMA were for this assessment. To guarantee the reliability of the
validation results, we selected stations that were close to the center of a target grid and
covered a long measurement period. Except for the ground-based observation data, the
RS SM data are available from the European Space Agency Water Cycle Multimission
Observation Strategy and Climate Change Initiative projects (ESA-CCI SM). The ESA-
CCI product provides relatively consistent and reliable information for SM worldwide
(Qiu et al., 2016) and has been successfully validated by many researchers (Dorigo et
al., 2015; Wang et al., 2016).




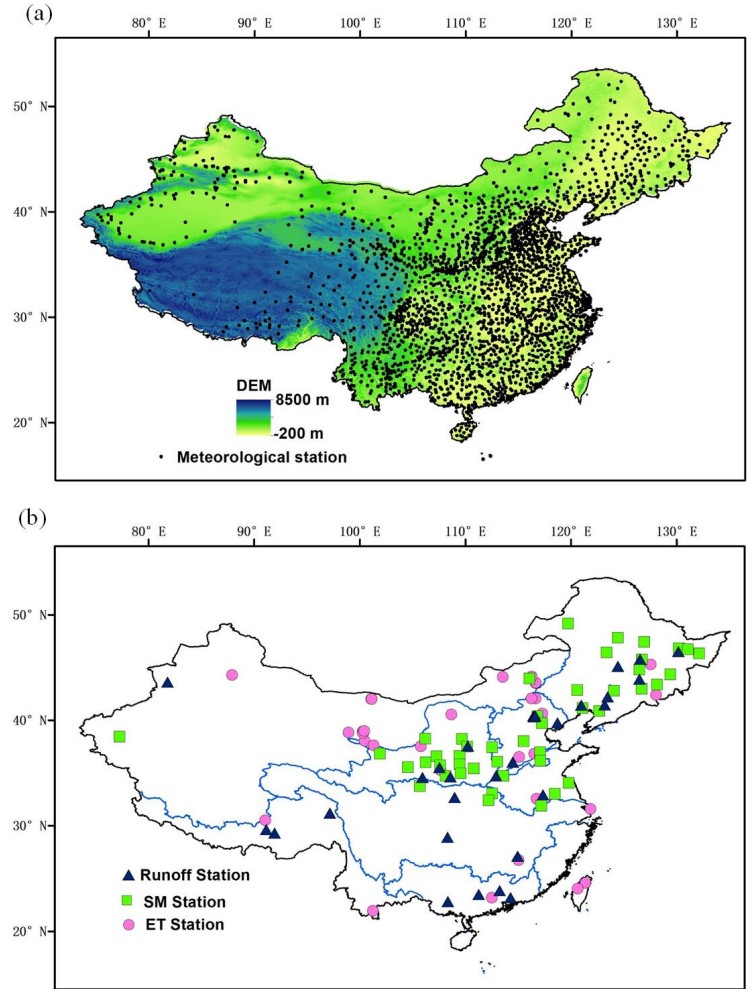


**Figure 1: Distribution of (a) meteorological stations and (b) in situ runoff**

**stations, SM and ET stations.**

**2.4 Parameter calibration and transfer scheme**
**2.4.1 Parameter calibration**
After all of the necessary input data for the model were collected and prepared, the VIC
model was calibrated for the selected 20 basins and validated for the 9 basins located
in different climate zones (Fig. 1). Most basins were minimally affected by human





activities, such as water extraction, irrigation, and water management. Seven of the
most sensitive VIC model parameters were targeted for calibration in each basin
separately, including the infiltration curve, $b$, the depths of the three soil layers
($d_1, d_2, d_3$), the maximum velocity of the baseflow, $D_{smax}$, the fraction of the
maximum baseflow velocity,  $D_s$, and the fraction of the baseflow of the maximum
SM where non-linear baseflow occurs, $W_s$. The parameters $D_{smax}$, $D_s$, $W_s$, and $d_3$ are
influential for runoff and for early season SM and ET since they govern water
infiltration and baseflow generation (Bennett et al., 2018). The initial values of these
sensitive parameters were obtained from Zhang et al. (2014) at a 0.25º resolution and
then were directly downscaled to a 0.0625º resolution. The calibration involved setting
an identical parameter set for each basin to find the best combination of the seven
parameters. It was performed via a trial and error procedure to match the simulations
with the hydrograph observations.
Three metrics were used to evaluate model performance: (1) the correlation coefficient
($R$), (2) the Nash-Sutcliffe efficiency (NSE), and (3) the relative error (bias; %) between
observations and simulations.
$\text{NSE} = 1 - \frac{\sum(Q_{i,obs} - Q_{i,sim})^2}{\sum(Q_{i,obs} - \overline{Q_{obs}})^2}$    (1)
$\text{Bias(\%)} = \frac{(\overline{Q_{sim}} - \overline{Q_{obs}})}{\overline{Q_{obs}}} \times 100\%$    (2)
In Eqs. 1 and 2, $Q_{i,obs}$ is the observed flow in the $i$ month, $Q_{i,sim}$ is the respective
$i$ th simulated flow from the model, and $\overline{Q_{sim}}$ and $\overline{Q_{obs}}$ are the observed and
simulated mean annual discharges for the calibration period, respectively.
For each grid cell in the calibrated basins, an adjustment factor (*Adj_factor*) can be





defined as:
$Adj\_factor = \dfrac{PAR_{final}}{\text{PAR}_{initial}}$ (3),
where $PAR_{final}$ and $PAR_{initial}$ are the final and the initial estimates of the parameter,
respectively. Based on this adjustment factor, the estimates of parameters in the
calibrated basins were transferred to the uncalibrated basins.
**2.4.2 Parameter transfer**
The area of China was divided into nine large river basins (Fig. 2) according to
topographic and LC conditions. As the VIC model parameters are closely related to
physical and climatic characteristics of basin properties, such as LC and meteorological
factors, we overlaid the river basins with climate zones to define a climatic similarity,
as described by Xie et al. (2007). Based on the climatic similarity and the adjustment
factor described in Sect. 2.4.1, the estimated parameters in calibrated basins were
transferred to the uncalibrated basins. The 20 independent, calibrated basins were
located in different climate zones and designed to estimate the parameters in their
uncalibrated, climate-related areas. Seven climatic zones in China, as defined based on
the Köppen classification criteria (Kottek et al., 2006), are shown in Table 1 and Fig. 2.
The parameter transfer strategy has been successfully used by Xie et al. (2007), and it
is briefly described as follows:
(1) The adjustment factors in each calibrated basin were used to adjust parameters in
uncalibrated basins;
(2) The rainy climate zone was further divided into three parts according to basins of
the Huai River, Yangtze River, and Pearl River, as C1, C2, and C3, respectively;



(3) The tropical climate zone has similar climatic characteristics to the Pearl River basin.
Therefore, the parameters for the tropical climate zone were set to the same adjustment
values as C3;
(4) Parameters of southeastern basins were used as the equivalent multiple as the
Yangtze River basin C2;
(5) The Dc climate zones covers two different regions in northeastern and southeastern
China (Dc east and Dc west). Therefore, the parameters in Dc east and Dc west were
adjusted using the same multiples from the related Da and E zones, respectively.
**Table 1: Classification of Köppen climate zones.**

| Climate zones | Description | Criterion |
|---|---|---|
| A | Equatorial climate | $T_{min} \geq +18$ ℃ |
| Bk | Dry, cold climate | $T_{ann} < +18$ ℃ |
| C | Rainy, midlatitude climate | $-3$ ℃ $< T_{min} < +18$ ℃ |
| Da | Continental climate with hot summer | $T_{max} \geq +22$ ℃ |
| Db | Continental climate with cool summer | $T_{min} \leq -3$ ℃ not (a) and at least 4 $T_{mon} \geq +10$ ℃ |
| Dc | Continental climate with short cool summer | not (Bk) and $T_{min} > -38$ ℃ |
| E | Polar climate | $T_{max} < +10$ ℃ |





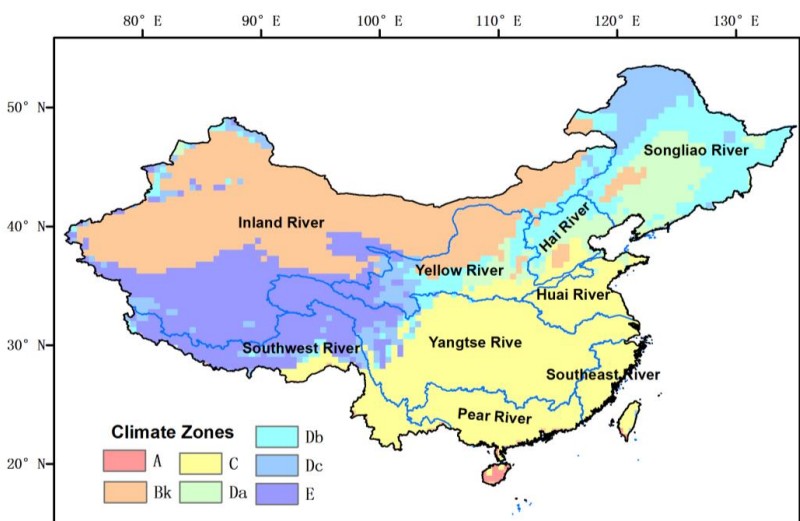


**Figure 2: River basins and climate zones in China.**

**3.   Results**
**3.1 Runoff calibration and validation**
To highlight the advantages of updating soil model parameters, we conducted two
simulations: one using the original soil parameters, which were directly downscaled
from a 0.25° resolution, and the other employing the updated soil parameters with
parameter calibration. Figure 3 presents the monthly discharge of the simulations from
the original and calibrated parameters and observations over 9 river basins, which were
chosen to be regionally representative and distributed among diverse climates. The
model performance was considerably better when using the calibrated parameters rather
than the initial parameters (Sect. 2.4.1). For most basins, the simulations with defaults
parameters tended to have higher discharges, especially overestimating the peak flow
during summer, such as in Phujym, Jilin, Heishiguan, and Tsuuang. In contrast, the
calibration was able to successfully avoid the overestimation of peak flow. However,



for the Shetang, Maojiahe, and Tsyamusy basins, which have little rainfall and runoff,
the initial parameters do not match the observations well at first during the low-flow
seasons, but this phenomenon changed after parameter calibration. Overall, the
comparisons revealed that the runoff dynamics were well captured after calibration, and
consequently the calibrated results were improved relative to the original VIC
simulations.
In Table 2, the model performances are listed for each basin after calibration and
validation. The correlation coefficient, NSE, and bias were used to evaluate the
simulations against observations. Most of the calibrated basins had high $R$ and NSE
values of more than 0.70. The relative bias presented here is generally within 20%. The
simulations of basins located in southern China (e.g., C1, C2, and C3), which usually
receive abundant rainfall and experience substantial runoff throughout the entire year,
tended to have better agreements with observations than those in northern China (e.g.,
Da, Db, and Bk). In general, the calibration improved the results in all instances,
although in some basins, such as Dingjiagou and Phujym, the results were still
unsatisfactory. A possible reason for such a discrepancy is that the VIC model is unable
to capture the impact of human activities, such as reservoir regulations.
The streamflow values simulated using the parameters sets through the parameter
transfer scheme were validated over 9 basins based on the observations. Compared with
the calibration process, six validation basins covering two climate zones, such as Da
and Db, were used to examine the performance of the parameter transfer approach.
Overall, the validation results (Table 2) were consistent with the calibration statistics.

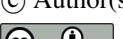



The $R$, NSE, and bias values for the validation period ranged from ~0.65–0.91, ~0.31–
0.87, and ~4.29–40.5%, respectively. The Zhangjiashan Basin had a relatively high bias,
mainly because there were only two years of observation data available for validation.
The best performance was found in the Hengshi Basin, while the worst was in the
Chiling Basin.

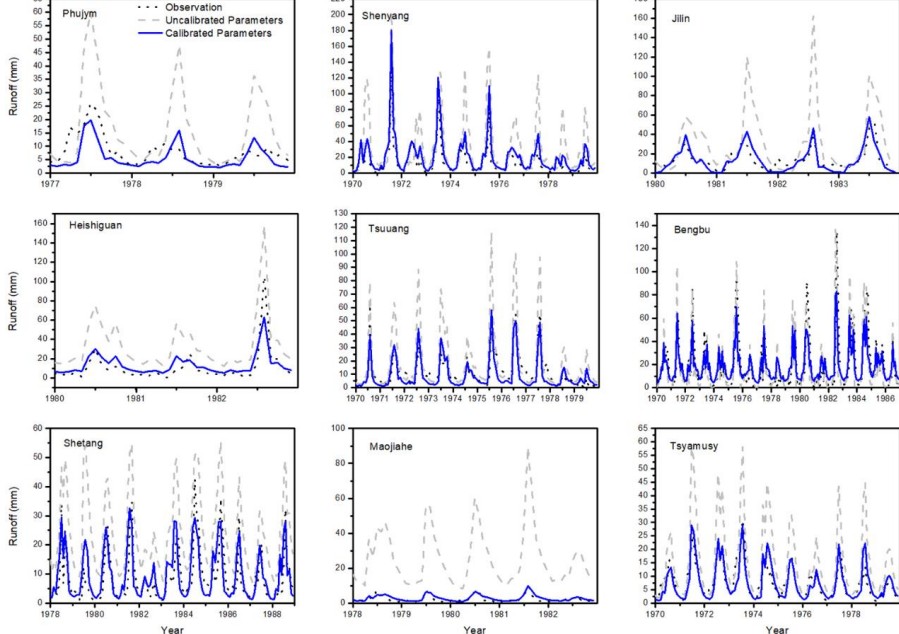


**Figure 3: Monthly discharges for some calibrated basins. The stippled lines are**
**observations, dashed lines are simulations from uncalibrated parameters, and**
**the solid lines are simulations from calibrated parameters.**

**Table 2: Statistics of calibrated and validated monthly flows.**

| Location | Latitude | Longitude | Climate Zone | Period | R | NSE | Bias |
|---|---|---|---|---|---|---|---|





| | Calibration | | | | | | |
|---|---|---|---|---|---|---|---|
| Yamadu | 43.62 | 81.8 | Bk | 2006-2008 | 0.91 | 0.59 | 7.59% |
| Dingjiagou | 37.55 | 110.25 | Bk | 1970-1986 | 0.47 | 0.26 | −20.70% |
| Bengbu | 32.93 | 117.38 | C1 | 1970-1986 | 0.82 | 0.65 | −2.78% |
| Tsuuang | 36.03 | 114.52 | C1 | 1970-1979 | 0.89 | 0.76 | −18.50% |
| Heishiguan | 34.71 | 112.93 | C1 | 1980-1982 | 0.91 | 0.72 | 25.40% |
| Jian | 27.1 | 114.98 | C2 | 1980-1982 | 0.86 | 0.75 | −4.86% |
| Ankang | 32.68 | 109.01 | C2 | 1980-1982 | 0.94 | 0.79 | 37.70% |
| Gongtan | 28.9 | 108.35 | C2 | 1980-1982 | 0.89 | 0.74 | −11.20% |
| Hoiyang | 23.17 | 114.3 | C3 | 1970-1982 | 0.92 | 0.74 | −3.54% |
| Wuzhou | 23.48 | 111.3 | C3 | 1970-1984 | 0.92 | 0.79 | 12.80% |
| Nanning | 22.8 | 108.36 | C3 | 1970-1983 | 0.87 | 0.74 | 13.60% |
| Shenyang | 41.46 | 123.24 | Da | 1970-1978 | 0.97 | 0.77 | 25.60% |
| Jilin | 43.88 | 126.53 | Da | 1980-1983 | 0.85 | 0.56 | −7.61% |
| Phujym | 45.1 | 124.49 | Da | 1977-1979 | 0.68 | 0.26 | 1.23% |
| Tsyamusy | 46.5 | 130.2 | Db | 1970-1978 | 0.86 | 0.69 | −3.84% |
| Shetang | 34.55 | 105.97 | Db | 1978-1988 | 0.78 | 0.58 | 12.40% |
| Maojiahe | 35.52 | 107.58 | Db | 1978-1982 | 0.84 | 0.61 | 28.60% |
| Yangcun | 29.3 | 91.96 | E | 1971-1975 | 0.88 | 0.6 | −6.77% |
| Changdu | 31.18 | 97.18 | E | 1975-1982 | 0.94 | 0.82 | 7.04% |
| Lasa | 29.63 | 91.15 | E | 1973-1975 | 0.93 | 0.81 | 5.26% |





| Validation | | | | | | | |
|---|---|---|---|---|---|---|---|
| Zhangjiashan | 34.63 | 108.60 | Bk, Db | 1980-1982 | 0.91 | 0.67 | 40.5% |
| Zhanjiafeng | 40.37 | 116.47 | Da, Db | 1970-1979 | 0.85 | 0.69 | 4.29% |
| Dalinghe | 41.41 | 121.00 | Da, Db | 1970-1979 | 0.91 | 0.76 | −6.97% |
| Chiling | 42.20 | 123.50 | Da, Db | 1970-1979 | 0.71 | 0.31 | 9.85% |
| Luanxian | 39.73 | 118.75 | Da, Db | 1970-1983 | 0.91 | 0.79 | 9.69% |
| Haerbin | 45.77 | 126.58 | Da, Dc | 1970-1983 | 0.78 | 0.51 | −9.87% |
| Hengshi | 23.85 | 113.27 | C3 | 1976-1979 | 0.95 | 0.87 | 15.5% |
| Qianxinzhuang | 40.32 | 116.55 | Db | 2006-2014 | 0.65 | 0.34 | 6.90% |
| Boyachang | 40.40 | 116.65 | Db | 2006-2014 | 0.84 | 0.68 | 9.76% |


**3.2 ET evaluation**
The root mean square error (RMSE) is a widely used measure of the differences
between model and observed variations (Yin et al., 2016). In this study, the RMSE was
also employed to estimate the differences between VIC model simulations and in-situ
observations. The statistics of the comparison between simulations and observations
are shown in Fig. 4 and Fig. 5 provides a comparison of some selected stations in the
four main basins (i.e., Songliao River Basin, Hai River Basin, Yellow River Basin and
Yangtse River Basin). The VIC model performed well and showed reasonable
consistency at the eddy covariance tower stations with respect to daily ET, with most $R$
values being greater than 0.6. The average RMSE values ranged between 0.6 mm and
3.6 mm. With respect to bias, many stations located in central China had values between



−60% and 20%. Weaker performances also occurred at a few stations mainly due to the
inconsistent scales of the two datasets, as the observation dataset includes single point
results, while model simulations are regionally averaged results. Therefore, the errors
may result from uncertainties in the in-situ measurements themselves and from
differences in the spatial scales between the model and the in situ measurements
(Gruber et al., 2013). As a whole, the strong relationships between ET simulations and
in-situ observations imply that they are qualitatively acceptable.
As for the spatial comparison of ET, Fig. 6 shows the seasonal changes and differences
between the VIC simulated ET and the GLASS ET. The VIC simulated larger ET values
in southeastern China and lower values for other areas relative to the GLASS products.
The differences ranged from −2 to 2 mm/day and this may have been caused by the
different temporal resolutions, which is 8 days for GLASS products. The average
difference for the four seasons was only approximately –0.36 mm, and thus the VIC
simulated ET was consistent with the RS estimated ET, implying an acceptable
performance for the model in this study.





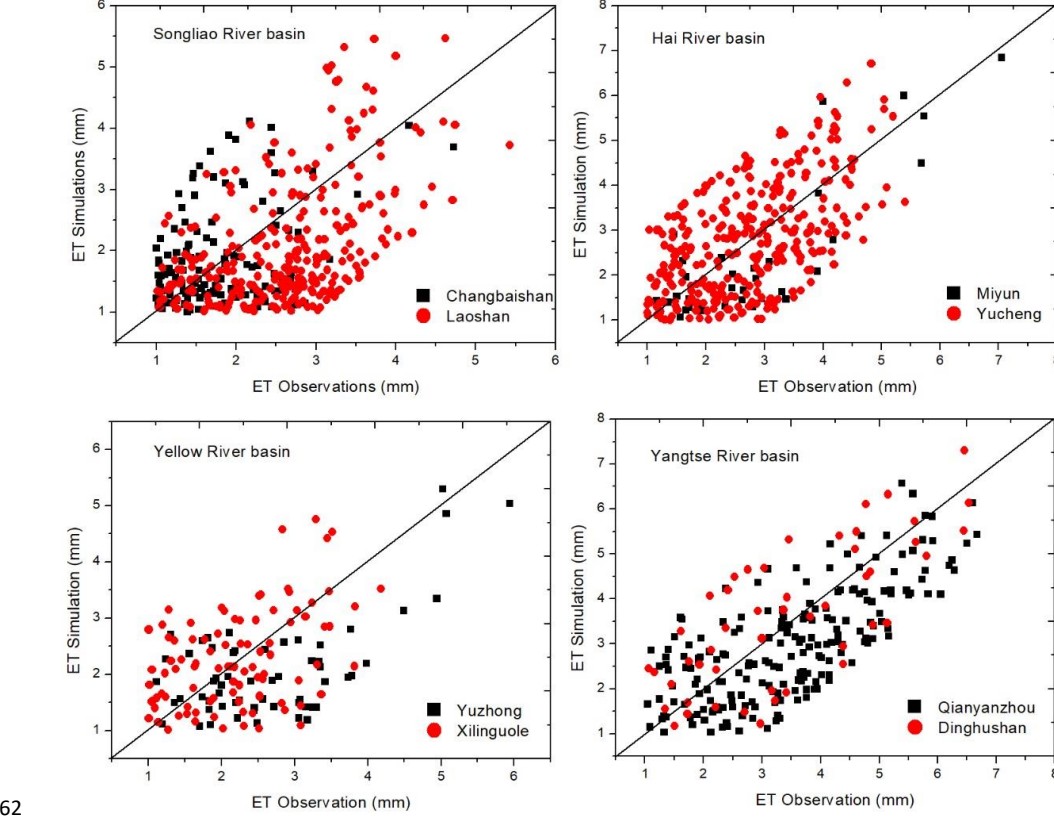


**Figure 4: Comparison of ET between observations and simulations of selected**
**stations in four basins.**





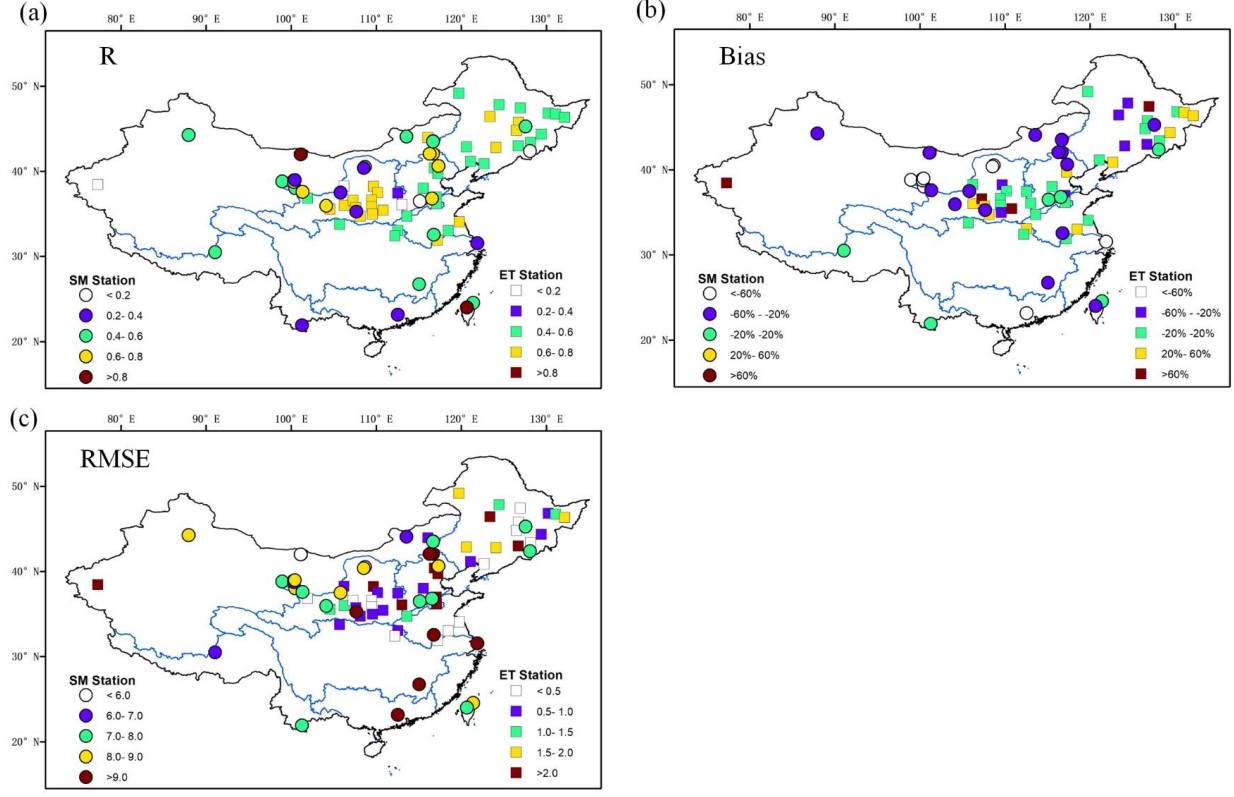

**Figure 5: Spatial distribution of the correlation coefficient (*R*), bias, and RMSE**

**between observations and simulations for ET and SM.**





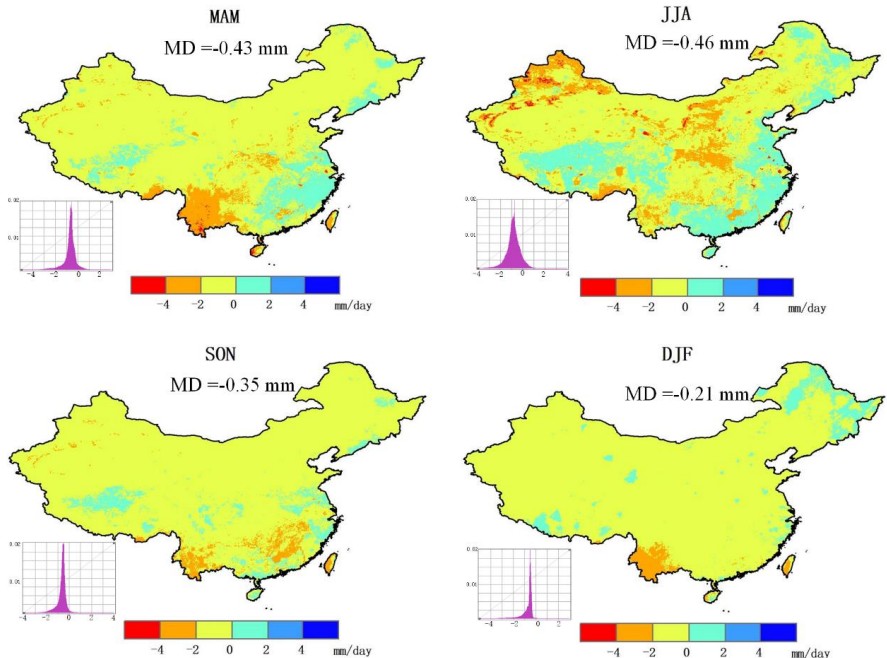

**Figure 6: Seasonal differences of ET between simulation and the GLASS for, from top to bottom, spring, summer, autumn, and winter. MD is the mean differences.**

**3.3 SM evaluation**

Figures 5 and 7 show the performance of the top 10-cm soil layer model estimates against in-situ SM observations. As shown in Fig. 5, the $R$ values for most stations were higher than 0.6 and the RMSEs were less than 12 mm. There was a pattern that emerged in which stations with a high $R$ values ($> 0.7$) usually showed considerably low RMSEs, indicating $R$ and the RMSE are not independent indicators of SM. Although several comparisons showed poor results, potentially because of the differences in temporal (10 days for observations and daily for simulations) and spatial resolutions. The stations located in central China, such as the Yellow River Basin and Hai River Basin, tended



to have lower biases, ranging from −20% to 20%. Meanwhile, the rural stations had
larger biases, which may due to limited and/or inaccurate observation data. Overall, the
comparisons of the two SM datasets demonstrated that they matched reasonably well.
The ESA-CCI SM product was used to evaluate the SM results from spatial perspective.
Figure 8 describes the differences between model simulations and ESA-CCI results.
The important differences in winter (December–February) can be seen in southwestern
China, and reached more than 10 mm. The opposite pattern appeared in summer (June–
August), with high differences exhibited in southeastern China. Slight differences
between simulations and the ESA-CCI product were found in spring (March–May),
with high values in the Yangtze River Basin. In autumn (September–November), the
southern region was covered by high differences values, which is distinctly different
from northern China. Combined, these results indicate that the VIC model with properly
calibrated parameters provides satisfactory simulations of runoff, ET, and SM.



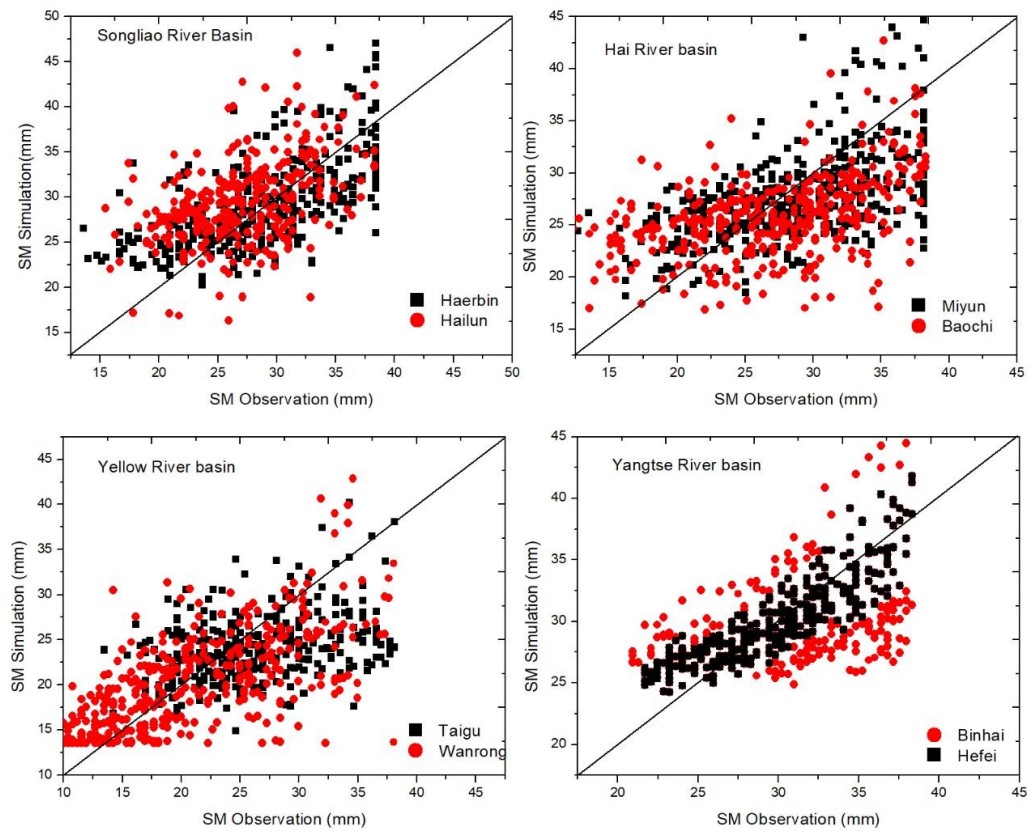


**Figure 7: Comparison between observations and simulations of selected stations**

396                                    **in four basins.**




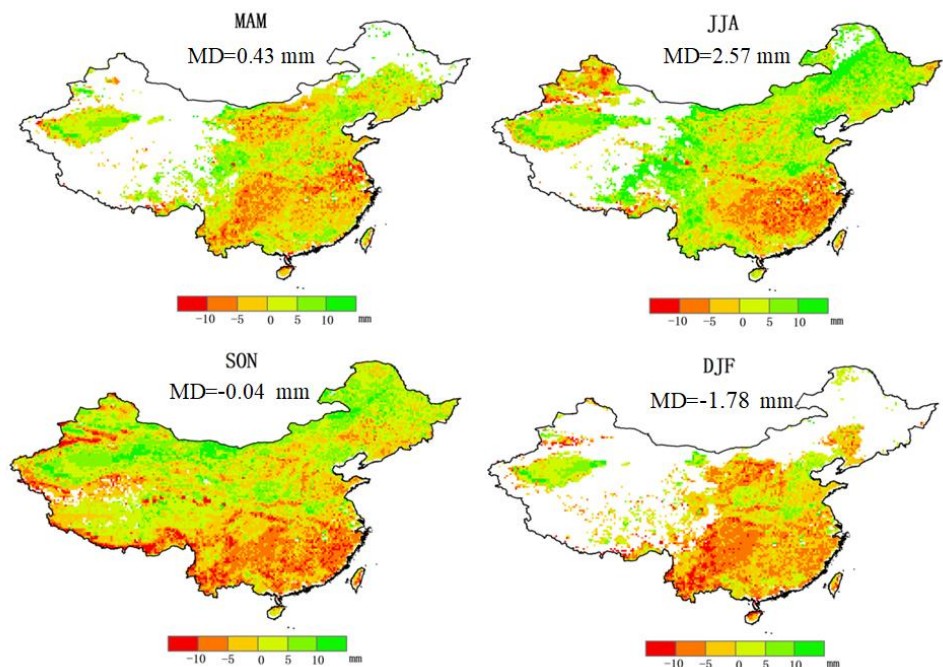


**Figure 8: Seasonal distributions of differences between simulations and the ESA-**

**CCI cases for the 0-10 cm soil layer SM. From top to bottom for: spring,**

**summer, autumn and winter. MD is the mean differences.**


**3.4 Application for detection of typical extreme events**

Flood and drought are the main natural disasters that occur in China, and they have
become important restricting factors for the development of society, the economy, and
agriculture (Zhang et al., 2016). However, the lack of high-resolution data makes
identifying flash floods and droughts over short timescales (pentads or weeks) nearly
impossible (Zhang et al., 2017). It is also necessary to monitor flood and drought events
in small regions, especially for remote areas without sufficient and reliable observation





data. Therefore, reliable, high-resolution modeling is essential for better analyzing
historical and predicted extreme events and then to make informed decisions in flood
and drought management. In this study, we applied the simulated 0.0625º dataset to
analyze two typical extreme disasters that occurred during recent years in China and to
evaluate the potential of the modeling to detect drought and flood events.
**3.4.1 Beijing flood event of 2012**
On July 21, 2012, the heaviest rainfall over the past six decades lashed Beijing. An area
of ~16,000 km$^2$ and more than 1.6 million people were affected by the flood (Wang et
al., 2013). A few studies have focus on the causes and patterns of this heavy rainfall
(Huang et al., 2014; Liu et al., 2003), while little attention has been paid to the
associated hydrological processes, such as the generation of runoff.
Here, we detected the flood coverage, which is represented by the runoff depth.
According to gauge observations, the intensive rainfall area extended from
southwestern Beijing to the northeastern areas (Chen et al., 2014). As shown in Fig. 9,
the runoff depth presented a SE–NE zonal distribution. However, the central region of
Beijing, which has the highest population and number of buildings, suffered the deepest
runoff, > 100 mm/day. This may have been due to the effects of urbanization during
recent years. There were four photos (Fig. 9) taken on July 21, 2012 that show the real
influence of this flood event (http://www.weather.cn).
To further evaluate the intensity of the flood event, we analyzed the frequency
distributions of precipitation and runoff (Fig. 10). The maximum precipitation was 287





mm over 24 hours, more than 50 mm of which was recognized as a rainstorm level in
76% of the area of Beijing. The mean precipitation is 103 mm on July 21, 2012.
Affected by the heavy rain, the maximum runoff is 172 mm; the average runoff is 26
mm for 24 hours. It should notice that the central urban area has the highest runoff
coefficient (Runoff/Precipitation) of 0.89, indicating that it will be at high flood risk
during urbanization when extreme rainfall happens (Wang et al., 2013).

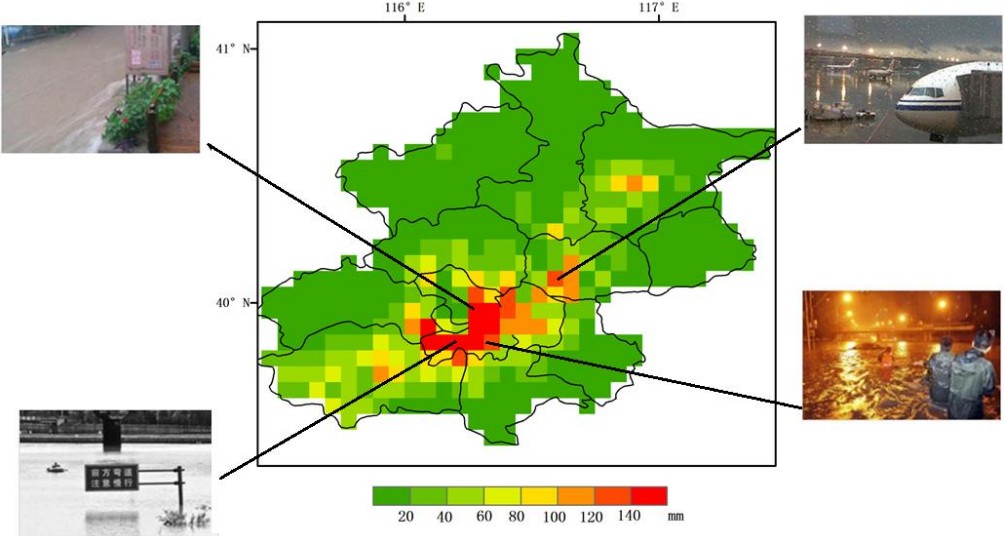


**Figure 9: Simulated runoff in Beijing on July 21, 2012**



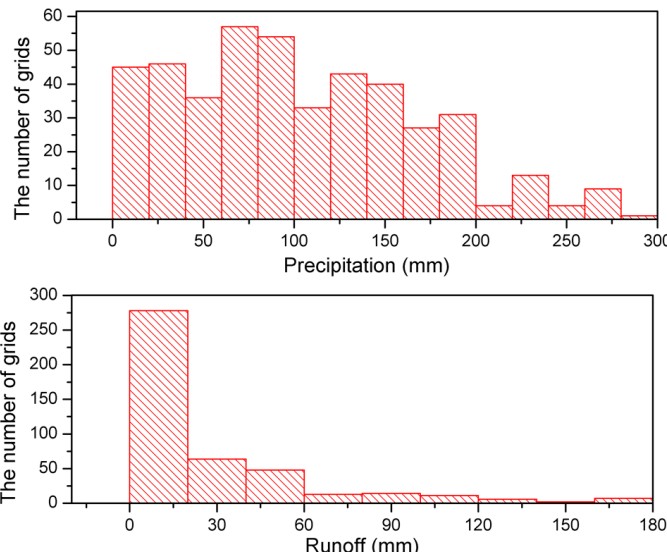


**Figure 10: Statistics of the numbers of grids with different precipitation and**

**runoff.**

**3.4.2 North China drought event of 2009**

From 2009–2010, a large-scale severe drought struck China (Ye et al., 2012). It lasted

for several months and subsequently has been considered as the most influential

drought event in northern and southwestern China (Zhu et al., 2018). In August 2009,

some portions of North China received only one inch of rainfall during the entire year.

In August of 2009, some portions of North China had received only one inch of rainfall

over the entire year.

As a result, this severe drought cost $100 million worth of losses. In this study, the VIC-

simulated SM was used to assess the severity and extent of the drought, particularly

focusing on agricultural drought, which was defined as a SM deficit. To emphasize the

advantage of high-resolution modeling for drought detection, we conducted a coarse-

resolution modeling at a 0.25º resolution that had the same sources of meteorological



forcing and soil and vegetation parameters as the 0.0625° modeling so that the only
difference between the two simulations was the spatial resolution.
The simulated drought event patterns of the two simulations showed a SE–NE zonal
distribution. However, differences between the simulations were obvious. The severe
drought in the 0.0625° simulations extended over more areas in northwestern and
southern China than in the 0.25° simulations. The Hai River Basin was selected in order
to distinguish the regional differences between the two simulations. The Hai River
Basin is one of the largest basins in North China, and contains a large population of 137
million people (Qin et al., 2015). It has long been identified as being sensitive to climate
change and has a recorded history of decades long droughts events. As shown in Fig.
11(c) and (d), the SM anomaly shows more detailed spatial structures with the increased
spatial resolution of the simulation.
In the 0.0625° simulations, drought mainly existed in the northwestern and northeastern
regions, and a few southwestern areas were also affected; these results are similar to
those of Wu et al. (2015) , which were based on RS data with a 1-km resolution.
However, the 0.25° simulations cannot show a detailed drought distribution.
Additionally, the magnitude of the SM anomaly in 0.0625° simulation results was larger
than in the 0.25° results, which ranged from −10.12 mm to 10.50 mm and −7.94 mm
to 4.75 mm, respectively (Fig. 11e). In the two simulations, 53.79% and 48.13% of
areas were affected by drought (i.e., the percentages of the SM anomaly were less than
zero), as shown in Fig. 11(f). These results indicate that the 0.0625° simulation could
successfully capture detailed spatial distributions and the severity of drought events.



Based on these analyses, we also used the 0.0625º simulations to analyze agricultural
drought events in the Hai River Basin over the last 46 years. Figure 12 shows the SM
anomalies and durations of the agricultural droughts, which were calculated by the
percentage of days with negative SM anomalies in a year from 1970 to 2015. As for the
SM anomalies, ~50% of past 46 years have experienced drought events. Severe drought
events occurred in 1972, the 1980s, 1999, and 2006. Moreover, there were 36 years in
which the droughts had relatively long durations (i.e., they lasted more than six months),
especially in 2006, when more than 75% of days had negative SM anomalies. Therefore,
the Hai River Basin is a typical drought-prone region. It suffered several intensive and
long-term drought events between 1980 and 1985, and in 1999 and 2006, and these
findings are consistent with the conclusions of Qin et al. (2015).





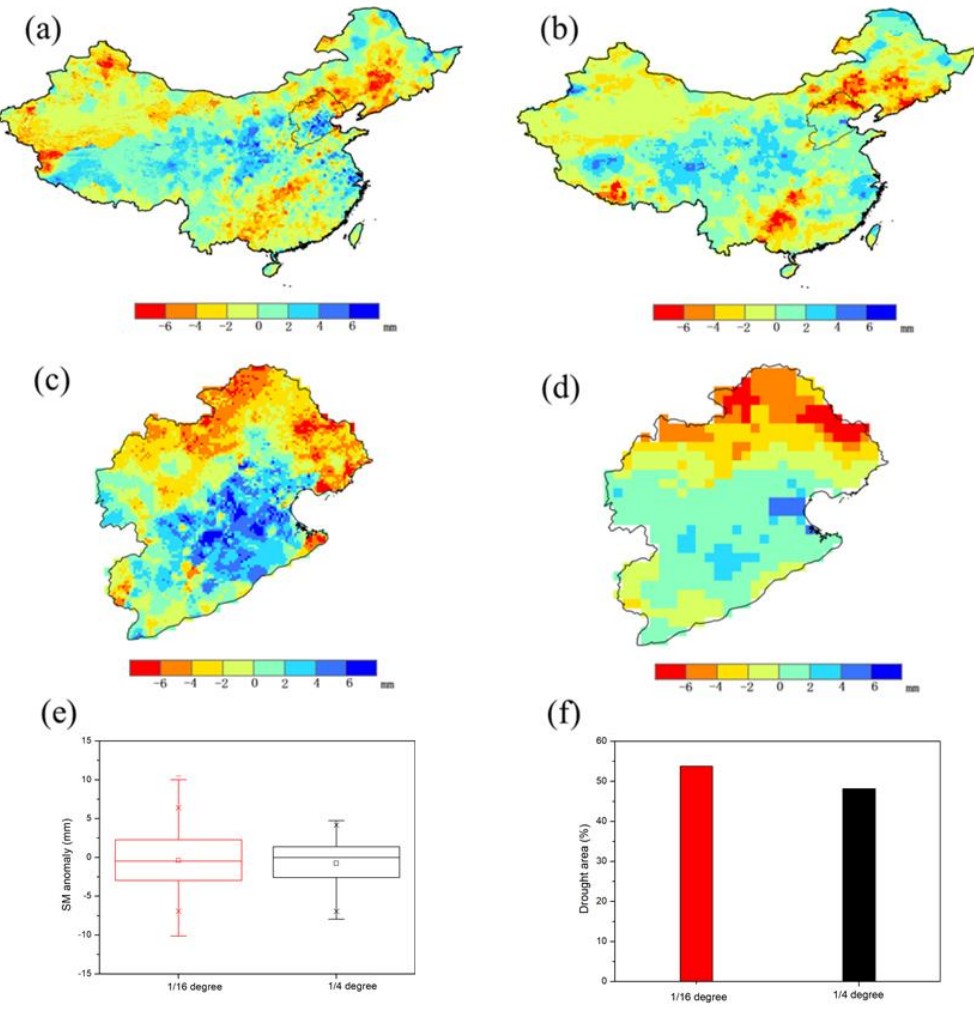


**Figure 11: Soil moisture anomaly from 0.0625º simulations in (a) China and (c)**

**the Hai River Basin, and from 0.25º simulations in (b) China and (d) the Hai**

**River Basin. The range of the (e) SM anomaly and (f) drought area of the Hai**

**River Basin from the two datasets are also shown.**





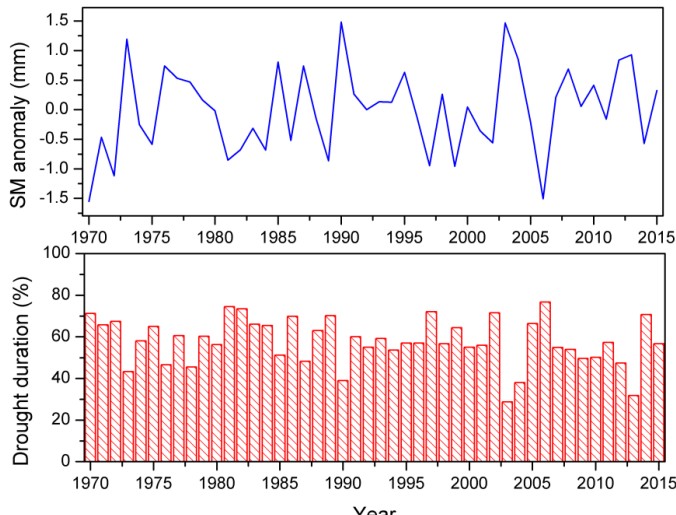

**Figure 12: Monthly SM anomaly and drought duration as a percentage of each**

**year from 1970 to 2015 in the Hai River Basin.**

**4. Discussion**

**4.1 Reliability of the modeling**

In this study, we developed a framework for the high-spatial resolution hydrological

modeling of runoff, ET, and SM at a 0.0625º spatial resolution across China from 1970–

2016. The model is highly reliable because, with respect to the forcing data, gridded

datasets were produced by interpolating station values from 2481 meteorological

stations across China, in contrast to the ~700 stations that have been commonly used in

many studies of China, such as that presented in the China Regional Surface

Meteorological Feature Dataset (CMFD). Meanwhile, high intensity station datasets

generally cover short periods of time, such as the CLDAS, which covered 2008–2016.

As for modeling process, a combination of climatic zones and river basin

methodologies was used to transfer the VIC parameters to uncalibrated areas. Soil



parameters, which are the most essential parameters of the VIC model, were adopted
from the newly released soil datasets of Dai et al. (2013) and Shangguan et al. (2013)
to improve the accuracy of hydrological modeling in our study. For model validation,
some previous studies have only validated runoff results using in-situ data (Lee et al.,
2017; Xie et al., 2007; Zhu and Lettenmaier, 2007), which may not guarantee the
reliability for other hydrological processes, such as ET and SM. In contrast, more
ground observations and RS data were used to validate simulations of runoff, ET, and
SM in our study.
Therefore, this high-quality, high-spatial resolution hydrological modeling could be
extended for relevant applications, such as detecting extreme events. As shown in Sect.
3.4, the simulations were capable of capturing detailed changes and providing reliable
information when drought and flood events occurred.
**4.2 Potential extension with China Land Data Simulation System (CLDAS) and**
**remote sensing (RS) data**
The CLDAS is a system that produces high-quality metrological forcing and SM
conditions over China at a 0.0625º resolution and in hourly time steps (Shi et al., 2011).
Three land surface models are included in the current version of the CLDAS-V2.0 (i.e.
CLM3.5, Noah-MP, and CoLM). In terms of the Global Land Data Assimilation System
(GLDAS) (Rodell et al., 2004) and the National Land Data Assimilation System
(NLDAS) (Mitchell, 2004), the VIC model is considered to fully simulate hydrological
processes.
In this study, the developed hydrological modeling framework based on VIC had the





same resolution as the CLDAS, and it was easy to couple with the CLDAS. Therefore,
this study provided an opportunity for the CLDAS to be combined with hydrological
modeling to better enhance its services.
Based on the high-quality and high-density drivers from the CLDAS, the simulation of
the VIC model could be applied to real-time hydrological process estimation across
China, and then offer an effective guide to detecting flood and drought events.
Furthermore, the RS data, such as LAI, albedo, and shortwave radiation, also could be
merged into the VIC, which may improve modeling results by considering the energy
balance.
**4.3 Limitations**
As shown in Sect. 3, the hydrological simulations were extensively validated with in
situ observations and RS data. However, with the exception of two stations, all of the
streamflow stations only had data records for the periods before 1990. The ET and SM
observations stations were mostly distributed in North China. Additionally, we
calibrated the most sensitive seven parameters of the VIC model $(b, d_1, d_2, d_3, D_{smax},$
$D_s$, and $W_s$), while the other parameters were not calibrated. For example, the
wintertime LAI and canopy fraction has a strong influence on variations in the snow
water equivalent (Bennett et al., 2018). Therefore, further efforts are needed to improve
model parameters uncertainties and the accuracy and application of RS products, and
to enhance the support of ground-based observation networks.
This study improved the spatial resolution of hydrological modeling to ~6 km across
China, which is just one step toward further increasing the resolution. The modeling



needs to be improved to reach a so-called hyper-resolution (~1 km or finer), which is
one of the "grand challenges" in current hydrological research (Wood et al., 2011).
Moreover, as hydrological processes generally evolve over various temporal scales,
from minute to daily time steps, future studies should also increase the evaluation of
temporal resolutions simultaneously (Melsen et al., 2016). However, the modeling in
our study was conducted roughly, at a daily time step, alone due to the limitations of
the forcing data. Hourly or smaller time step data can capture more detailed processes,
such as flash floods, infiltration, and pore flow (Blöschl and Sivapalan, 1995).
Furthermore, the achievement of high-spatial and temporal-resolution modeling not
only requires the resolution to increase, but also involves the development of
hydrological models to consider hydrological processes that are consistent with such
high resolutions, including lateral groundwater flow (Zeng et al., 2018; Zeng et al.,
2016) and efficient runoff routing algorithms (Li et al., 2013; Meng et al., 2017; Wen
et al., 2012; Wu et al., 2014).
**5. Conclusion**
In order to address the fundamental questions associated with the effects of
environmental changes across various scales, we developed a high-resolution (0.0625º)
hydrological modeling for China using the VIC model over the period from 1970–June

569   2016.

The modeled runoff, ET, and SM were fully calibrated and validated against the data
from in-situ stations and RS. The modeled runoff results were significantly improved
after parameter calibration and transfer using a combination of climatic zones and river



basin methodologies. Additionally, the *R* and NSE values of most calibrated and
validated basins were greater than 0.70, and the relative bias was generally below 20%.
The simulations of humid regions, such as the Yangtze River Basin, tended agree better
with observations than those of arid regions. Furthermore, ET and SM simulations were
also validated against ground observations and RS products. The *R* and RMSE values
for ET and SM were quite acceptable. The simulated ET and SM and the RS products
(e.g., GLASS, ESA-CCI) were consistent across spatial and temporal distributions.
Therefore, the hydrological modeling is capable of capturing the hydrological processes
at such a high resolutions, and can provide reliable estimates of land surface
hydrological conditions in China.
Several important implications emerge from our work. For example, this
implementation has a higher spatial resolution and generally improved performance
relative to earlier model results (Lee et al., 2017; Zhang et al., 2014; Zhu and
Lettenmaier, 2007). The increased spatial resolution improves the ability of the
modeling to represent topographic effects and resolve smaller watersheds, and hence
provide information relevant to local water management concerns, such as on drought
and flood events.
Consequently, this is the first time that hydrological states and fluxes at a 0.0625º spatial
resolution have been produced for China, and they are freely available to analyze multi-
scale hydrological, ecological, and meteorological interactions and initial conditions.
Additional efforts will be needed to improve the hydrological modeling by using more
detailed model inputs and advanced parameter calibration techniques. Moreover, there



is great potential for the extension of our modeling results with CLDAS and RS data to
improve high-resolution modeling applications.
**Acknowledgements**
This study was supported by grants from the National Key Research and Development
Program of China (NO.2016YFA0600103, No. 2016YFC0401404) and the National
Natural Science Foundation of China (No. 41471019).

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
