# Peer review of "Toward high-spatial resolution hydrological modeling"

_Hydrology and Earth System Sciences, 2019_

## Referee Comment (RC1) · Anonymous Referee #1 · 30 Mar 2019

This study implemented a high-resolution (1/16) hydrological modeling over China based on the Variable Capacity (VIC) hydrologic model, wherein the VIC parameters was calibrated with the streamflow data record from 29 gauging stations. Comparing with the available in-situ/satellite-based products, the validation analyses demonstrated that the calibrated VIC hydrological modeling at a 0.0625° spatial resolution is overall able to reproduce the key water budget terms, including the runoff hydrographs, evapotranspiration (ET) patterns, and soil moisture (SM) dynamics. The results may benefit the VIC model to be coupled with the operational China Land Data Assimilation System (CLDAS). Although this manuscript is well written and of good readability, I do have a few concerns to be addressed.

1. The form of this manuscript is very reminiscent of past work by others. A general

comment is that the authors need to clearly highlight the unique of such high-resolution off-line modeling dataset comparing with the existing similar datasets, including global coverage. 2. Some assertions about model performance are made arbitrarily due to lacking of authoritative criteria. For instance, in terms of evaluating model calibration, the authors can cite one reference (Moriasi et al., 2007, doi:10.13031/2013.23153) that places a lower range to describe a "satisfactory" calibration. 3. Typically, the hydrological model is calibrated with long-term (>10-yr at least) streamflow observation record and validated over another independent period. In the current version, however, the record length of most calibration stations (Table 2) is too short (less than 3-yr) to ensure the robustness of model performance. Also, the streamflow validation over an independent period is still lacking for each calibration station. 4. Soil moisture (SM) memory play an important role in the land surface water and energy budget. The authors should add the evaluation with respect to the SM persistency. 5. VIC outputs include a set of snow related files, which are important for water and energy balance in the cold or mountainous regions. Please add the validation analysis of VIC snow output. 6. Line 240-242. Please provide more details on the parameter interpolation. 7. Line 282-283. "southeastern China" should be "southwest China". 8. Line 340-343. This sentence is subject to grammar mistake. Please double-check this issue. 9. The quantitative metric information is absent in most of figures. For instance, please add the RMSE information in each panel of Figure 4 and Figure 7. 10. Figure 3 presents the comparison of monthly discharge, but the Y-axis is labeled with runoff (mm), rather than with discharge (m3/s). Please address this issue.

---

## Referee Comment (RC2) · Anonymous Referee #2 · 10 Apr 2019

This study presents a long term high-resolution hydrological modeling over China by using a hydrological model called VIC. The VIC parameters were estimated by using high resolution land surface data (e.g., soil texture, LAI and land use category) and a manual calibration procedure to my understanding. A 6-km simulation was validated against in-situ streamflow, ET and soil moisture observations, as well as remote sensing products. While high-resolution modeling is important for addressing regional phenomena and fine-scale processes, this manuscript fails to present advances or advantages of such effort. Except for using an existing or widely used 1-km land surface data, there is no update on VIC model physical processes specifically for high-resolution simulation (e.g., lateral flow, urban model). It is not clear whether the 6-km simulation improves the modeling of water cycle. The English presentation also needs

extensive edits. Therefore, I have to recommend for a rejection. A few comments are listed below.

Major comments: 1. Providing local relevantly information is the most important features of high resolution modeling. However, the current work does not provide the evidence for the improvement of high resolution against coarse resolution simulation. A comparison is needed to compare the 6-km simulation of streamflow, soil moisture and ET with those from a coarse resolution (e.g., 25 or 30 km). To make a fair comparison, the coarse resolution model should also be calibrated. Then, the advantages or add values of high-resolution hydrological modeling could be illustrated. This is one of the most important issues of the current manuscript.

2. Another major concern is whether VIC modeling outperforms other land surface models. This could be answered by comparing the VIC simulation with existing reanalysis data, including GLDAS, CLDAS etc. This could demonstrate whether current study does provide solid advances in high-resolution modeling.

3. The third major concern is whether the water-balance VIC model without any updates in representing local human interventions (e.g., reservoir operation, irrigation, groundwater pumping, urbanization) is valid at high resolution. If these local human-relevant phenomena are totally ignored, the science rationale of high-resolution modeling is questionable.

4. The English should be improved, perhaps with help from a native speaker. Many presentations are not professional for an international journal.

Minor comments: 5. L244-245: "It was performed via a trial and error procedure to match the simulations with the hydrograph observations." As you have seven parameters, they would have too many combinations to calibrated manually. Which measure did you use to decide to stop the trial?

6. In section 3.1, the author tends to show the advantages of updating soil parameters

by comparing a simulation which uses coarse resolution soil parameters with those by using the high resolution results. But I think the obvious improvement at high resolution may be directly due to the calibration as you do not calibrate the model at coarse resolution.

7. Figure 1 shows about half of the soil moisture observation stations are located over the Yellow River basin, where the human influence is large (such as irrigation). Although the author says "Most basins were minimally affected by human activities, such as water extraction, irrigation, and water management", the human water intervention cannot be ignored over the Yellow River, which is heavily managed.

8. The Beijing flood event of 2012 was used to show model ability of capturing flood events, but can the VIC model represent the urbanization effects on hydrological regimes? As shown in Line 425-426, "However, the central region of Beijing, which has the highest population and number of buildings, suffered the deepest runoff, > 100 mm/day. This may have been due to the effects of urbanization during recent years." How is the urbanization represented in the model, by changing land use category or using other method? To my understanding, the urban model is missing in the publicly-used version of the VIC model.

9. The definition of flood and drought. The author used runoff depth anomaly and soil moisture anomaly to represent flood and drought respectively, but the detail definition is not given. Which anomaly threshold is taken as flood and drought?

10. Line 187-190. The soil dataset provided by Dai et al., (2013) has 10 layers which is the same as Common Land Model, how do you match the 10 layers to the 3 layers in VIC model, by simple average or other methods?

11. Line 230-232. Please give the information of which basins you used for the calibration and validation, as there are only 9 basins in Figure 1.

12. Figure 4 and Figure 7, Please give the R-square and p value of the results.

13. Section 4.3 As shown above, if you calibrated model manually, the limitations of the work should consider this. There are some automatically calibration method to find the optimal parameter combination, which will improve your streamflow simulations.

14. Line 423: "the runoff depth presented a SE–NE zonal distribution", it seems to be "SW-NE" instead of "SE-NE".

15. L37, L113 and many other places in the manuscript, CLDAS means CMA Land Data Assimilation System instead of "China Land Data Simulation System".

16. L79. This is not true, many studies started to use the meteorological observations based on 2K+ stations.

17. L252, "observed and simulated" -> "simulated and observed"

18. Table 2. The streamflow calibration and validation results are not that favorable. Many coarse resolution simulations could have higher NSE values. A more robust calibration procedure is needed.

19. Figures 4, 5 and 7. The validation results should be compared with those from un-calibrated model, calibrated model at coarse resolution, as well as the state-of-the-art land reanalysis data.

20. Figure 9. This does not make sense since the version of VIC used in this study does not have urban component.

---

## Author Comment (AC1) · 10 Apr 2019

This study implemented a high-resolution (1/16) hydrological modeling over China based on the Variable Capacity (VIC) hydrologic model, wherein the VIC parameters was calibrated with the streamflow data record from 29 gauging stations. Comparing with the available in-situ/satellite-based products, the validation analyses demonstrated that the calibrated VIC hydrological modeling at a 0.0625_ spatial resolution is overall able to reproduce the key water budget terms, including the runoff hydrographs, evapotranspiration (ET) patterns, and soil moisture (SM) dynamics. The results may benefit the VIC model to be coupled with the operational China Land Data Assimilation System (CLDAS). Although this manuscript is well written and of good readability, I do have a few concerns to be addressed.

Reply: Thanks for the constructive comments. Please see the below response point-by-point.

1. The form of this manuscript is very reminiscent of past work by others. A general comment is that the authors need to clearly highlight the unique of such high-resolution off-line modeling dataset comparing with the existing similar datasets, including global coverage.

Reply: The purpose of this study is to develop a high-resolution hydrological modeling over China with a resolution of 1/16th degree and to show a potential to couple with the operational CLDAS. The modeling also provided hydrological dataset at this resolution. There are several features in the dataset. First, it holds high-spatial resolution, while existing simulated hydrological datasets for China have a coarse resolution, such as 1/4th degree (Zhang et al., 2014). The 1/16th degree simulations including ET, Runoff and SM could present more detailed information for detection of flooding and drought events (shown in section 3.4). Second, the dataset follows a physical constrain with energy and water balance that are well defined in the land surface hydrological model. In contrast, satellite remote sensing products generally have a limitation regarding the physical constrain despite their high resolution. And third, the dataset in this study was extensively evaluated using ground-based observations and remote sensing products.

2. Some assertions about model performance are made arbitrarily due to lacking of authoritative criteria. For instance, in terms of evaluating model calibration, the authors can cite one reference (Moriasi et al., 2007, doi:10.13031/2013.23153) that places a lower range to describe a "satisfactory" calibration.

Reply: Thanks for your suggestion. Moriasi et al. (2007) provided a summary for the statistics in model evaluation (e.g., NSE and PBIAS). As to the runoff simulation, our modelling presents a favorable performance regards of NSE comparing with the statistic median NSE which is 0.6. We will cite this reference in the revision.

3. Typically, the hydrological model is calibrated with long-term (>10-yr at least) streamflow observation record and validated over another independent period. In the current version, however, the record length of most calibration stations (Table 2) is too short (less than 3-yr) to ensure the robustness of model performance. Also, the streamflow validation over an independent period is still lacking for each calibration station.

Reply: We agree that the model evaluation at a few stations with relative short length of streamflow records may not assure the robustness of model performance. We discussed this limitation in section 4.3 and expect that more observations are available in future. The simulated hydrograph represents a natural flux without considering human activities. Streamflow data from a few stations may be affected by reservoir regulation or irrigation (Wang et al., 2013), which are not suitable for model validation because the current version of VIC fails to characterize such human activities. To remedy the limitation in streamflow evaluation, we employed soil moisture and ET data from in-situ observations and remote sensing products in order to evaluate the model performance. This further evaluation presents a favorable performance, indicating the VIC modeling in this study is robust for the state and fluxes simulation.

4. Soil moisture (SM) memory play an important role in the land surface water and energy budget. The authors should add the evaluation with respect to the SM persistency.

Reply: Thanks for suggestion. As for SM persistency, the autocorrelation of simulated SM is calculated as a function of the monthly lag of three selected stations, shown in Figure 1. The time-scales of simulated SM memory is 1-2 month which is similar to (Entin et al., 2000). Here we just present a simple calculation of SM memory as a few other factors may affect SM persistency (Hagemann and Stacke, 2014). We will provide more evolutions on SM memory in the revision.

5. VIC outputs include a set of snow related files, which are important for water and energy balance in the cold or mountainous regions. Please add the validation analysis of VIC snow output.

Reply: Thanks for suggestion. The high-resolution hydrological modeling discussed in the manuscript mainly including Runoff, ET and SM. As for snow cover, which is also an essential part of hydrological cycle in cold regions, has not been fully focused in our study due to uncertainties of VIC model in snow cover simulations (Islam and Déry, 2017). In the revision, we will validate the simulated snow cover dataset with the remote sensing product.

6. Line 240-242. Please provide more details on the parameter interpolation.

Reply: Each 1/4th degree grid cell contains 16 sub-grid cells of 1/16th degree resolution. Therefore, we regard each sub-grid cell has the same parameters as 1/4th degree grid. However, as for soil hydraulic properties parameters, (i.e., field capacity, wilting point, saturated hydraulic conductivity, and bulk density) for each of the three layers were obtained from the soil dataset (Dai et al., 2013) and then prescribed to the 1/16th grid in this study. These sensitive parameters of 1/16th degree have been fully calibrated after interpolation.

7. Line 282-283. "southeastern China" should be "southwest China".

Reply: Thanks. We will revise the words as suggested.

8. Line 340-343. This sentence is subject to grammar mistake. Please double-check this issue.

Reply: Thanks for your comment. We will modify this sentence in the revised manuscript.

9. The quantitative metric information is absent in most of figures. For instance, please add the RMSE information in each panel of Figure 4 and Figure 7.

Reply: The RMSE information between simulations and observations for SM and ET are represented in Figure 5 (c), which show the spatial distribution of RMSE. We will add the RMSE value in the Figure 4 and Figure 7 for the comparisons between observations and simulations of selected stations.

10. Figure 3 presents the comparison of monthly discharge, but the Y-axis is labeled with runoff (mm), rather than with discharge (m3/s). Please address this issue.

Reply: The runoff simulated by VIC represents the depth of runoff water, which unit is mm. To keep the consistence of two dataset, we converted the unit of observations (m3/s) to runoff (mm) during calibration and validation. We will address this issue in our manuscript.

Reference: Dai, Y., Shangguan, W., Duan, Q., Liu, B., Fu, S., and Niu, G.: Development of a China Dataset of Soil Hydraulic Parameters Using Pedotransfer Functions for Land Surface Modeling, Journal of Hydrometeorology, 14, 869-887, 10.1175/jhm-d-12-0149.1, 2013.

Entin, J. K., Robock, A., Vinnikov, K. Y., Hollinger, S. E., Liu, S., and Namkhai, A.: Temporal and spatial scales of observed soil moisture variations in the extratropics, Journal of Geophysical Research: Atmospheres, 105, 11865-11877, 2000.

Hagemann, S., and Stacke, T.: Impact of the soil hydrology scheme on simulated soil moisture memory, Climate Dynamics, 44, 1731-1750, 10.1007/s00382-014-2221-6, 2014. Islam, S. U., and Déry, S. J.: Evaluating uncertainties in modelling the snow hydrology of the Fraser River Basin, British Columbia, Canada, Hydrology and Earth System Sciences, 21, 1827-1847, 10.5194/hess-21-1827-2017, 2017.

Moriasi, D. N., Arnold, J. G., Liew, M. W. V., Bingner, R. L., Harmel, R. D., and Veith, T. L.: Model Evaluation Guidelines for Systematic Quantification of Accuracy in Watershed Simulations, Transactions of the ASABE, 50, 885-900, 10.13031/2013.23153, 2007.

Wang, W., Shao, Q., Yang, T., Peng, S., Xing, W., Sun, F., and Luo, Y.: Quantitative assessment of the impact of climate variability and human activities on runoff changes: a case study in four catchments of the Haihe River basin, China, Hydrological Processes, 27, 1158-1174, 10.1002/hyp.9299, 2013.

[Figure]

Zhang, X.-J., Tang, Q., Pan, M., and Tang, Y.: A Long-Term Land Surface Hydrologic Fluxes and States Dataset for China, Journal of Hydrometeorology, 15, 2067-2084, 10.1175/jhm-d-13-0170.1, 2014.

Please also note the supplement to this comment:
https://www.hydrol-earth-syst-sci-discuss.net/hess-2019-72/hess-2019-72-AC1-supplement.pdf

————————————

[Figure]

Fig. 1. Figure 1 Autocorrelation of simulated SM as a function of the monthly lag of three selected stations.

---

## Author Comment (AC2) · 16 Apr 2019

Anonymous Referee #2 This study presents a long term high-resolution hydrological modeling over China by using a hydrological model called VIC. The VIC parameters were estimated by using high resolution land surface data (e.g., soil texture, LAI and land use category) and a manual calibration procedure to my understanding. A 6-km simulation was validated against in-situ streamflow, ET and soil moisture observations, as well as remote sensing products. While high-resolution modeling is important for addressing regional phenomena and fine-scale processes, this manuscript fails to present advances or advantages of such effort. Except for using an existing or widely used 1-km land surface data, there is no update on VIC model physical processes specifically for high resolution simulation (e.g., lateral flow, urban model). It is not clear whether

the 6-km simulation improves the modeling of water cycle. The English presentation also needs extensive edits. Therefore, I have to recommend for a rejection. A few comments are listed below.

Reply: We thank the referee for the valuable comments. We do agree that improving model physical processes is important for a high resolution simulation. Charactering fine-scale physical processes (e.g., agricultural irrigation, lateral flow, and urbanization effect) is still a great challenge in most of land surface hydrological models. Instead of improving the physical processes, the aim of this study is to calibrate the VIC model for China at the 6-km resolution. After this evaluation, the model is expected to be coupled with the CLDAS. Our study makes a step forward for a country-scale modeling with a relatively high resolution. For the simulation in this study, we will provide more evaluations and comparisons using reanalysis data (e.g., data from CLDAS and GLDAS).

Major comments: 1. Providing local relevantly information is the most important features of high resolution modeling. However, the current work does not provide the evidence for the improvement of high resolution against coarse resolution simulation. A comparison is needed to compare the 6-km simulation of streamflow, soil moisture and ET with those from a coarse resolution (e.g., 25 or 30 km). To make a fair comparison, the coarse resolution model should also be calibrated. Then, the advantages or add values of high-resolution hydrological modeling could be illustrated. This is one of the most important issues of the current manuscript.

Reply: We presented a comparison between the 1/16th-degree simulation and the 1/4th-degree simulation in section 3.4.2. The results indicated that the high resolution simulation is able to provide more reliable and detailed information over small areas and sub-basins. Please note the coarse resolution modeling has been calibrated by Zhang et al. (2014).

2. Another major concern is whether VIC modeling outperforms other land surface

models. This could be answered by comparing the VIC simulation with existing reanalysis data, including GLDAS, CLDAS etc. This could demonstrate whether current study does provide solid advances in high-resolution modeling.

Reply: This is a good idea. We will add the evaluation by comparing the simulations with reanalysis data, such as GLDAS and CLDAS to illustrate the advantages of our simulations.

3. The third major concern is whether the water-balance VIC model without any updates in representing local human interventions (e.g., reservoir operation, irrigation, groundwater pumping, urbanization) is valid at high resolution. If these local human relevant phenomena are totally ignored, the science rationale of high-resolution modeling is questionable.

Reply: We agree that these human activities have certain influence on hydrological processes. Besides the human interventions, there are many natural factors (e.g., water and vegetation interaction, baseflow formulation, preferential flow) that may also impose great impact on a high-resolution modeling. These limitations exist in most of hydrological models (Zhu and Lettenmaier, 2007;Lee et al., 2017;Zhang et al., 2014) and land surface data assimilation systems (Rodell et al., 2004;Shi et al., 2011;Charusombat et al., 2012), including the CLDAS which has the same resolution as this study. Therefore, we argue currently there are challenges in high resolution modeling regarding fine-scale processes. This study intends to calibrate the VIC model over China at the resolution as CLDAS. So this step is referred to as a "toward high resolution modeling". After this model calibration, we will attempt to improve the physical processes for high-resolution modeling by including such human activities and natural processes into the VIC model.

4. The English should be improved, perhaps with help from a native speaker. Many presentations are not professional for an international journal

Reply: We will improve the language of manuscript.

Minor comments: 5. L244-245: "It was performed via a trial and error procedure to match the simulations with the hydrograph observations." As you have seven parameters, they would have too many combinations to calibrated manually. Which measure did you use to decide to stop the trial?

Reply: The seven parameters show different influences on the model performance. The infiltration parameter (bi) and the second soil depth (d2) are intensively calibrated for surface runoff, while the three baseflow parameters and the third soil layer (d3) have only minor adjustment. Please note the seven parameters have been well calibrated by Zhang et al. (2014) at 0.25-degree resolution. Based on the parameter values from Zhang et al. (2014), we tried different combinations of bi and d2 , and calculated the measures of model performance (e.g. the Nash-Sutcliffe efficiency, NSE). The calculation will stop if NSE gives negligible improvement. After this calibration, the three baseflow parameters and d3 undergo minor adjustments to improve model performance.

6. In section 3.1, the author tends to show the advantages of updating soil parameters by comparing a simulation which uses coarse resolution soil parameters with those by using the high resolution results. But I think the obvious improvement at high resolution may be directly due to the calibration as you do not calibrate the model at coarse resolution.

Reply: We are sorry that our presentation may make the referee misunderstand the coarse-resolution simulation. The soil parameters for the coarse-resolution simulation (the 1/4th degree modeling) have been well calibrated by Zhang et al. (2014) in which the simulations of streamflow are consistent with observations.

7. Figure 1 shows about half of the soil moisture observation stations are located over the Yellow River basin, where the human influence is large (such as irrigation). Although the author says "Most basins were minimally affected by human activities, such as water extraction, irrigation, and water management", the human water intervention

cannot be ignored over the Yellow River, which is heavily managed.

Reply: We agree that the ground-based observations of soil moisture located over the Yellow River basin may be affected by human activities. So the model renders relatively poor performance (the correlation coefficient R is within 0.2-0.4). However, it's the most available observation dataset for validation over China. To remedy the limitation, we employed soil moisture from remote sensing products in order to evaluate the model. This further evaluation presents a favorable performance, indicating the VIC modeling in this study is robust for the state and fluxes simulation.

8. The Beijing flood event of 2012 was used to show model ability of capturing flood events, but can the VIC model represent the urbanization effects on hydrological regimes? As shown in Line 425-426, "However, the central region of Beijing, which has the highest population and number of buildings, suffered the deepest runoff, > 100 mm/day. This may have been due to the effects of urbanization during recent years." How is the urbanization represented in the model, by changing land use category or using other method? To my understanding, the urban model is missing in the publicly used version of the VIC model.

Reply: Thanks for your comment. Although the current version of VIC cannot represent the urbanization effects, the urbanization can be partly represented by the land use characteristics such as LAI, FVC and albedo in model input file. We will revise this description in the manuscript to reduce misunderstanding.

9. The definition of flood and drought. The author used runoff depth anomaly and soil moisture anomaly to represent flood and drought respectively, but the detail definition is not given. Which anomaly threshold is taken as flood and drought?

Reply: The purpose of section 3.4 is to show the advantage of the high resolution modeling by comparing the simulations with different resolutions. The drought event was simply defined as the case with negative soil moisture anomaly, and the flood event was defined as the case when surface runoff larger than long-term average. The

definitions will be specifically presented in the revised paper.

10. Line 187-190. The soil dataset provided by Dai et al., (2013) has 10 layers which is the same as Common Land Model, how do you match the 10 layers to the 3 layers in VIC model, by simple average or other methods?

Reply: We averaged soil hydraulic data from Dai et al. (2013)for the corresponding layers. For example, for the top layer of the VIC model (with the thickness of 10 cm), we averaged soil hydraulic data of the multiple layers of which the total thickness is equal to 10 cm. The same method was used to obtain soil parameters for the other two layers.

11. Line 230-232. Please give the information of which basins you used for the calibration and validation, as there are only 9 basins in Figure 1.

Reply: Please see the Table 2 for the information of calibration and validation stations.

12. Figure 4 and Figure 7, Please give the R-square and p value of the results.

Reply: We will provide more statistics information of the results.

13. Section 4.3 As shown above, if you calibrated model manually, the limitations of the work should consider this. There are some automatically calibration method to find the optimal parameter combination, which will improve your streamflow simulations.

Reply: Thanks. We will add this suggestion into limitation.

14. Line 423: "the runoff depth presented a SE–NE zonal distribution", it seems to be "SW-NE" instead of "SE-NE".

Reply: Will be revised as suggested.

15. L37, L113 and many other places in the manuscript, CLDAS means CMA Land Data Assimilation System instead of "China Land Data Simulation System".

Reply: Will be revised as suggested.

16. L79. This is not true, many studies started to use the meteorological observations based on 2K+ stations.

Reply: We will reword this sentence in manuscript.

17. L252, "observed and simulated" -> "simulated and observed"

Reply: We will revise this in manuscript.

18. Table 2. The streamflow calibration and validation results are not that favorable. Many coarse resolution simulations could have higher NSE values. A more robust calibration procedure is needed.

Reply: We will revise this in manuscript.

19. Figures 4, 5 and 7. The validation results should be compared with those from uncalibrated model, calibrated model at coarse resolution, as well as the state-of-the-art land reanalysis data.

Reply: We will add more comparisons with reanalysis data in manuscript.

20. Figure 9. This does not make sense since the version of VIC used in this study does not have urban component.

Reply: The urban component in the current VIC version can be partly represented by land cover type and land characteristics. Although the model does not fully couple with urban process, the simulations could show a relative realistic runoff generation process in urban area. This section will be improved to avoid confusion.

Reference:

Charusombat, U., Niyogi, D., Garrigues, S., Olioso, A., Marloie, O., Barlage, M., Chen, F., Ek, M., Wang, X., and Wu, Z.: Noah-GEM and Land Data Assimilation System (LDAS) based downscaling of global reanalysis surface fields: Evaluations using observations from a CarboEurope agricultural site, Computers and Electronics in Agriculture, 86, 55-74, 10.1016/j.compag.2011.12.001, 2012.

Dai, Y., Shangguan, W., Duan, Q., Liu, B., Fu, S., and Niu, G.: Development of a China Dataset of Soil Hydraulic Parameters Using Pedotransfer Functions for Land Surface Modeling, Journal of Hydrometeorology, 14, 869-887, 10.1175/jhm-d-12-0149.1, 2013.

Lee, K., Gao, H., Huang, M., Sheffield, J., and Shi, X.: Development and Application of Improved Long-Term Datasets of Surface Hydrology for Texas, Advances in Meteorology, 2017, 1-13, 10.1155/2017/8485130, 2017.

Rodell, M., P.R. Houser, U. Jambor, J. Gottschalck, K. Mitchell, C.-J. Meng, K. Arsenault, B. Cosgrove, J. Radakovich, M. Bosilovich, J.K. Entin, J.P. Walker, D. Lohmann, and Toll, D.: The Global Land Data Assimilation System, Bull. Amer. Meteor. Soc., 85, 381-394, 2004.

Shi, C. X., Xie, Z. H., Hui, Q., Liang, M. L., and Yang, X. C.: China land soil moisture EnKF data assimilation based on satellite remote sensing data, Science China Earth Sciences, 54, 1430-1440, 2011.

Zhang, X., Q. Tang, M. Pan, and Tang, Y.: A Long-Term Land Surface Hydrologic Fluxes and States Dataset for China, Journal of Hydrometeorology, 10.1175/JHM-D-13-0170.1, 2014.

Zhu, C., and Lettenmaier, D. P.: Long-Term Climate and Derived Surface Hydrology and Energy Flux Data for Mexico: 1925–2004, Journal of Climate, 20, 1936-1946, 10.1175/jcli4086.1, 2007.

Please also note the supplement to this comment:
https://www.hydrol-earth-syst-sci-discuss.net/hess-2019-72/hess-2019-72-AC2-supplement.pdf